# The Role of the Estimated Plasma Volume Variation in Assessing Decongestion in Patients with Acute Decompensated Heart Failure

**DOI:** 10.3390/biomedicines13010088

**Published:** 2025-01-02

**Authors:** Andreea-Maria Grigore, Mihai Grigore, Ana-Maria Balahura, Gabriela Uscoiu, Ioana Verde, Camelia Nicolae, Elisabeta Bădilă, Adriana-Mihaela Ilieșiu

**Affiliations:** 1Department Cardio-Thoracic Pathology, Carol Davila University of Medicine and Pharmacy, 020021 Bucharest, Romania; andreea-maria.banciu@drd.umfcd.ro (A.-M.G.); ana-maria.balahura@umfcd.ro (A.-M.B.); gabriela.uscoiu@umfcd.ro (G.U.); ioana.verde@umfcd.ro (I.V.); dr_camelia_nicolae@yahoo.com (C.N.); elisabeta.badila@umfcd.ro (E.B.); adriana.iliesiu@umfcd.ro (A.-M.I.); 2Cardiology Department, Colentina Clinical Hospital, 020125 Bucharest, Romania; 3Internal Medicine and Cardiology Department, “Prof. Th. Burghele” Clinical Hospital, 050653 Bucharest, Romania

**Keywords:** heart failure, decongestion, plasma volume

## Abstract

Introduction and Aim: Assessing decongestion in patients with acute decompensated heart failure (ADHF) is challenging, requiring multiple parameters and often remaining imprecise. The study aimed to investigate the utility of indirectly estimating plasma variation (∆ePVS) for evaluating decongestion in ADHF patients in relation to natriuretic peptides. Materials and Methods: This prospective, observational, single-center study included 111 patients (mean age 74 years, 40% female) hospitalized with ADHF and treated with intravenous diuretics along with optimized medical therapy. Patients were clinically and echocardiographically evaluated at admission, with blood tests performed at both admission and discharge. A decrease of ≥30% in NT-proBNP at discharge was considered a marker of decongestion. ∆ePVS was calculated using the Strauss formula: ∆ePVS (%) = 100 × [(Hb admission/Hb discharge) × (1 − Hct discharge)/(1 − Hct admission)] − 100. A negative ∆ePVS (<0%) at discharge was considered a marker of hemoconcentration. Patients were divided into two groups: G1 (∆ePVS < 0%, 81 patients) and G2 (∆ePVS ≥ 0%, 30 patients). Results: Both groups had similar left ventricular ejection fraction (LVEF) values of 46%, mean hemoglobin (Hb) (12 g/dL), and creatinine (1.16 ± 0.65 mg/dL). NT-proBNP decreased in 88% patients in G1 and in 26% patients in G2 (*p* < 0.001). During hospitalization, five patients from G2 died. At 6 months, rehospitalization occurred in 35% of G2 and 21% of G1 (*p* = 0.04), with mortality rates of 37% in G2 and 11% in G1 (*p* = 0.012). Multivariate regression identified ∆ePVS as the only significant predictor of NT-proBNP decrease (OR 0.11, 95% CI 0.04–0.33, *p* < 0.001). Conclusions: Indirect estimation of plasma volume and its variation are valuable, accessible, and cost-effective parameters for assessing decongestive treatment in ADHF patients, complementing natriuretic peptides.

## 1. Introduction

Congestion is the primary cause of hospitalization in patients ADHF [1,2]. It is often underdiagnosed, as detecting congestion remains challenging because its clinical manifestations represent only the final stage of evolving pathophysiological processes [3,4,5].

The importance of evaluating decongestion stems from its role as one of the most significant negative predictors, influencing the prognosis of heart failure (HF) patients through early rehospitalizations and increased mortality [6]. In patients hospitalized with ADHF, a careful evaluation of congestion before discharge is recommended [7].

However, relying solely on clinical signs may lead to inadequate patient management. Currently, there are several methods for assessing decongestion, such as the ultrasound evaluation of the venous flow in the portal vein or intrarenal veins, but these methods can be time-consuming and require a steep learning curve [8]. Plasma volume variation during decongestive therapy has been proposed as a parameter to monitor a patient’s progress toward euvolemia [9].

The gold standard for measuring plasma volume is the radioisotope assay; however, this method is expensive and impractical, as it requires multiple venous blood samples [10].

Non-invasive equations have been validated as alternative methods for ∆ePVS [9]. These equations are based on the assumption that variations in Hb concentrations over time are inversely related to changes in intravascular total blood volume (TBV). Based on these beliefs, changes in Hb concentration are expected to be inversely proportional to changes in TBV [11]. However, there is limited evidence regarding the use of plasma volume to predict outcomes in ADHF, and most studies analyzing ∆ePVS have been retrospective, primarily including patients with chronic HF, with only a few prospective studies focusing on patients with ADHF [9,11].

Several formulas are available for ∆ePVS, but the Strauss formula was the only that has been validated against a radiolabeled gold-standard method and has been consistently used for decades to estimate plasma volume in patients undergoing plasma exchanges [12,13].

The aim of this study is to evaluate ∆ePVS for monitoring decongestion in ADHF patients and to assess its prognostic role.

The study endpoints include in-hospital mortality, six-month mortality, and hospitalization for ADHF.

## 2. Materials and Methods

This prospective, observational study was conducted at a single center from 1 November 2022 to 1 January 2023. All patients provided informed consent, and the study adhered to the ethical standards outlined in the Declaration of Helsinki (1964) and its subsequent amendments as well as national guidelines for good medical practice. The study received approval from the Hospital’s Ethics Committee.

The diagnosis of ADHF was established based on the 2021 European Society of Cardiology guidelines for HF [7].

Inclusion criteria included the following:Patients with episodes of decompensated HF, regardless of left ventricular ejection fraction (LVEF).

Exclusion criteria included the following:
Severe anemia (Hb < 8 g/dL);Worsening anemia, defined as a decrease in Hb > 2 g/dL during hospitalization;Patients having received a blood transfusion, specifically those with an increase in Hb of more than 3 g/dL from admission to discharge;Hyponatremia requiring correction;Advanced malignancies with a life expectancy of less than 1 year;Septic shock;Patients with stage V chronic kidney disease (eGFR < 15 mL/min/1.73 m^2^) or hemodialysis;Any type of surgical intervention;Pregnant females.

All patients received intravenous loop diuretics within the first 48 h of admission, with dose adjustments based on clinical and biochemical response. Monitoring included daily weight measurement, renal function tests, and NT-proBNP levels at admission and discharge.

Assessments were conducted at two time points: the first within the first 24 h of admission and the second on the day of discharge.

### 2.1. Clinical and Laboratory Evaluation

Clinical evaluations were performed at admission, and lower-limb edema was assessed clinically by applying digital pressure to evaluate pitting and swelling. Laboratory analyses were conducted at both admission and discharge, including Hb, Ht, serum creatinine, and estimated glomerular filtration rate (eGFR), calculated using the CKD-EPI formula, as well as serum urea, serum sodium (Na), serum potassium (K), and NT-proBNP. Worsening renal function (WRF) was defined as an increase in creatinine of more than 0.3 mg/dL between the two time points [14]. A significant decrease in NT-proBNP was defined as a reduction ≥ 30% from the initial value [15].

Estimation of Plasma Volume Variation

Multiple formulas are available for estimating plasma volume; however, the Strauss formula was selected due to its validation against a radiolabeled gold-standard method and its long-standing use over decades in estimating plasma volume for patients undergoing plasma exchanges [12,13].

The definitions of hemoconcentration and the Strauss formula are based on the assumption that variations in Hb concentrations over time are inversely related to changes in intravascular TBV. Based on these beliefs, changes in Hb concentration are expected to be inversely proportional to changes in TBV [11].

The Strauss formula is as follows:$$∆ePVS=100\times \frac{hemoglobin\left(\frac{g}{dL}\right)admission}{hemoglobin\left(\frac{g}{dL}\right)discharge}\times \frac{100-hematocrit\left(\%\right)admission}{100-hematocrit\left(\%\right)discharge}-100$$

Based on ∆ePVS, patients were divided into two groups:Group 1 (<0%)—hemoconcentration;Group 2 (≥0%)—hemodilution.

∆ePVS was chosen for its validated use as a surrogate for plasma volume changes, offering a non-invasive, cost-effective, and widely applicable method based on routine clinical parameters.

### 2.2. Ultrasound

Echocardiography was performed at admission using a GE Vivid E95 Ultrasound Machine (GE HealthCare, Chicago, IL, USA) with a 1.4–4.6 MHz transducer and simultaneous ECG recording. Several parameters were measured at admission according to the latest European Association of Cardiovascular Imaging recommendations [16], including LVEF, left atrial diameter (LAd), and estimation of E/e′ filling pressures. Right-heart parameters such as tricuspid annular plane systolic excursion (TAPSE), lateral tricuspid annulus peak systolic velocity (S′), systolic pulmonary artery pressure (sPAP), and inferior vena cava diameter (IVCd)/collapse for sPAP estimation were also recorded [16]. Images were analyzed and processed using the offline EchoPAC PC v204 software.

Lung ultrasound was performed using a GE Vivid E95 ultrasound machine equipped with a 6.0–13.0 MHz transducer to identify pulmonary congestion, defined by the presence of a minimum of three B-lines in two zones of a hemithorax, and to assess for pleural effusion, which appears as an anechoic structure located between the pleura and the lung parenchyma [17].

Discharge

Discharge was based on clinical stability, including symptom resolution, improved biochemical parameters, and echocardiographic parameters, as determined by the attending physician.

Follow-up

Patient follow-up was conducted either via telephone or by reviewing hospital records at 180 days.

ADHF hospitalization was defined as an admission where HF was the primary diagnosis, characterized by symptoms and signs of HF that necessitated treatment with intravenous diuretics. We used the same criteria for “HF” as we did for ADHF in the inclusion criteria. Information on these endpoints was gathered through a review of records in local hospital databases, telephone interviews with patients, and consultation of documents from other hospitals.

### 2.3. Statistical Analysis

The statistical analysis was performed using IBM SPSS software (version 20 for Windows). Descriptive statistics were calculated for all patients. For continuous variables, data are presented as the mean with standard deviation for normally distributed data or as the median and interquartile range (IQR) for non-normally distributed data. The Shapiro–Wilk test was used to assess the normality of the data distribution. Categorical variables are summarized as absolute numbers and percentages.

The relationship between quantitative variables was evaluated using either Pearson’s or Spearman’s correlation tests, depending on the data distribution, while associations between categorical variables were analyzed with the chi-square test. For within-group comparisons over the two time points for non-normally distributed variables, the Friedman test was applied. Comparisons between groups were carried out using the Student’s *t*-test for normally distributed variables and the Mann–Whitney test for non-normally distributed ones.

Receiver operating characteristic (ROC) curve analysis was employed to assess the predictive value of quantitative variables. To identify independent predictors of mortality and readmission, multivariate regression analysis was performed, with results expressed as odds ratios (OR) and 95% confidence intervals (CI).

A *p*-value of less than 0.05 was considered statistically significant for all tests.

## 3. Results

The patient selection process for the study is detailed in Figure 1. Initially, 131 patients with ADHF were identified; however, 15 patients were excluded due to severe anemia at admission. An additional five patients were excluded because of worsening anemia during hospitalization. As a result, a total of 111 patients were included in the final analysis.

A total of 111 patients were divided at discharge based on ∆ePVS into two groups: Group 1 with 81 patients and Group 2 with 30 patients. The median age was 73 years in Group 1 and 74 years in Group 2, with no statistically significant difference between the groups (*p* = 0.20) (Table 1). The proportion of female patients was similar, accounting for 40.7% in Group 1 and 36.67% in Group 2 (*p* = 0.56). Body mass index (BMI) was also comparable between the groups, with a median value of 28.57 kg/m^2^ in Group 1 and 30.51 kg/m^2^ in Group 2 (*p* = 0.23) (Table 1).

Comorbidities

Ischemic heart disease was present in 44.44% of patients in Group 1 and 26.67% in Group 2, although this difference was not statistically significant (*p* = 0.141). HTN was observed in 81.48% of Group 1 and 86.67% of Group 2 (*p* = 0.49). DM affected 39.51% of Group 1 and 30% of Group 2 (*p* = 0.39). CKD was more prevalent in Group 2 (73.33%) compared to Group 1 (56.79%), but this did not reach statistical significance (*p* = 0.169). AF was present in 54.32% of Group 1 and 66.67% of Group 2 (*p* = 0.30). COPD was similar in both groups (20.99% in Group 1 vs. 26.67% in Group 2, *p* = 0.49) (Table 1).

Medication at admission

Regarding medication at admission, the use of ACEi was similar between the groups (32.10% in Group 1 vs. 36.67% in Group 2, *p* = 0.54), as was the use of ARB (18.52% in Group 1 vs. 10% in Group 2, *p* = 0.33). ARNI usage was observed in 20.99% of Group 1 and 10% of Group 2 (*p* = 0.24) (Table 1).

The use of beta-blockers at admission was higher in Group 1 (90.12%) compared to Group 2 (70%, *p* = 0.06), although this difference was not statistically significant.

The use of MRA was also significantly higher in Group 1 (79.01%) versus Group 2 (43.33%, *p* = 0.0008). No significant differences were noted in the use of SGLT-2 inhibitors (37.04% in Group 1 vs. 30% in Group 2, *p* = 0.47) (Table 1).

### 3.1. Clinical Congestion Parameters

The assessment of congestion parameters across all 111 patients yielded several findings, as presented in Table 2.

Regarding weight, the mean at admission for the entire cohort was 83 kg (±18.02). Group 1 reported a mean weight of 83.5 ± 18.82 kg, while Group 2’s mean weight was 83.8 ± 15.95 kg, indicating no significant difference between the groups (*p* = 0.92). At discharge, the mean weight for the overall group decreased to 82 kg (±17.10), with Group 1 showing a mean weight of 80 ± 17.77 kg and Group 2 showing 82 ± 14.55 kg; again, no statistically significant difference was observed (*p* = 0.17).

In terms of lower-limb edema, it was present in 36% of the total cohort at admission, with 31% in Group 1 and 48% in Group 2, but this difference was not statistically significant (*p* = 0.28). By discharge, the prevalence of edema had decreased to 9% overall, with Group 1 showing 7% and Group 2 showing 12%, which also did not reach statistical significance (*p* = 0.15).

Jugular vein distension was observed in 71% of the entire cohort at admission, with Group 1 reporting a similar prevalence of 68% and Group 2 at 75% (*p* = 0.38). At discharge, the overall prevalence of JVD decreased to 13%, with 11% in Group 1 and 15% in Group 2, again showing no significant difference (*p* = 0.29).

### 3.2. Echocardiography and Lung Ultrasound

Ultrasound evaluations, including echocardiography and lung ultrasound, were performed for all 111 patients (Table 3).

The echocardiographic findings revealed a mean LVEF of 46.74% ± 15.08 for the entire cohort. In Group 1, the mean LVEF was 46.19% ± 15.16, while Group 2 had a mean LVEF of 46.03% ± 12.77, indicating no significant difference between the groups (*p* = 0.95). The median E/e′ medial ratio was 28 (IQR 20–37) for the total cohort, with Group 1 showing a median of 28.33 (IQR 19–39) and Group 2 showing 29 (IQR 21–32), which was not statistically significant (*p* = 0.31).

Regarding the LAd, the mean was 47 mm ± 6.99 across all patients, with Group 1 reporting a mean of 47 mm ± 5.8 and Group 2 reporting 48 mm ± 8.9, revealing no significant difference (*p* = 0.87). The TAPSE averaged 17 mm ± 3.9 for the overall cohort, with Group 1 and Group 2 presenting means of 17 mm ± 3.8 and 16 mm ± 4.2, respectively, also showing no significant difference (*p* = 0.50).

The mean S′RV for the total cohort was 11 cm/s ± 2, with Group 1 at 11.73 cm/s ± 1.9 and Group 2 at 11.31 cm/s ± 2.5, again indicating no significant difference (*p* = 0.49). The diameter of the IVC averaged 22 mm ± 4.8 for all patients, with Group 1 measuring 22 mm ± 4.6 and Group 2 measuring 21 mm ± 5.3, which was not statistically significant (*p* = 0.82). The mean sPAP was found to be 49 mmHg ± 14.1 overall, with Group 1 and Group 2 showing means of 49 mmHg ± 13 and 50 mmHg ± 16, respectively (*p* = 0.36).

In terms of lung ultrasound findings, pleural effusion was present in 48.65% of the total cohort, with 46.91% in Group 1 and 53.33% in Group 2, reflecting no significant difference (*p* = 0.45). Signs of pulmonary congestion were noted in 87.39% of the overall patients, with a prevalence of 86.42% in Group 1 and 90% in Group 2, which also did not show a significant difference (*p* = 0.54).

### 3.3. Biochemical Parameters and Diuretic Treatment

Biochemical and diuretic treatment data were collected at both admission and discharge for all 111 patients (see Table 4).

Regarding Hb levels, the mean at admission for the entire cohort was 12 ± 2.66 g/dL. In Group 1, the mean Hb level was slightly higher at 12.3 ± 2.68 g/dL, while Group 2 presented a lower mean of 11.75 ± 2.63 g/dL, with no statistically significant difference observed (*p* = 0.90). At discharge, the overall mean Hb level increased to 12.7 ± 2.4 g/dL. Group 1 exhibited a further increase to 13.4 ± 2.34 g/dL, whereas Group 2’s mean Hb level decreased to 11.2 ± 2.74 g/dL, again showing no significant difference (*p* = 0.33).

In terms of Ht, the mean at admission for the total population was recorded at 37.4 ± 7.7%. Group 1 had a mean Ht of 37.34 ± 7.62%, while Group 2 reported a mean of 37.8 ± 7.82%, indicating no significant difference (*p* = 0.92). At discharge, the overall mean Ht increased to 39.1 ± 7.02%. Group 1’s mean was higher at 40.6 ± 6.79%, while Group 2’s mean Ht decreased to 34.5 ± 9.01%, with no statistically significant difference between the groups (*p* = 0.33).

The median furosemide dose at admission was 20 mg/day (IQR 0–40 mg/day) for both groups, with no significant difference between them (*p* = 0.06). At discharge, the median dose increased to 60 mg/day (IQR 40–80 mg/day) for both groups, with Group 2 receiving a higher dose (*p* = 0.03).

The median total furosemide dose during hospitalization was 85 mg/day (IQR 60–110 mg/day) across both groups, with a lower dose in Group 1 [75 mg/day (IQR 50–90 mg/day)] compared to Group 2 [95 mg/day (IQR 70–120 mg/day)].

### 3.4. ∆ePVS and Renal Function

At admission, the mean creatinine level was significantly different between the groups, with an overall level of 1.16 ± 0.65 mg/dL. Group 1 had a mean creatinine level of 1.08 ± 0.49 mg/dL, while Group 2 reported a higher mean of 1.34 ± 0.57 mg/dL (*p* = 0.048). By discharge, the overall mean creatinine level decreased to 1.19 ± 0.43 mg/dL, with Group 1 showing a level of 1.13 ± 0.49 mg/dL and Group 2 at 1.32 ± 0.61 mg/dL (*p* = 0.00418).

Patients in Group 2, with worsening renal function in contrast to Group 1 without WRF, had a significantly higher risk of in-hospital mortality (OR 1.66, 95% CI 1.10–2.51, *p* = 0.02) and six-month mortality (OR 1.87, 95% CI 1.16–3.01, *p* < 0.01). However, they did not significantly predict the risk of rehospitalization (OR 1.28, 95% CI 0.76–2.15, *p* = 0.28).

The mean serum urea level at admission was significantly higher in Group 2, with a value of 73 ± 55 mg/dL compared to 54 ± 30 mg/dL in Group 1 (*p* = 0.0051). At discharge, the overall mean serum urea level was 59 ± 37.1 mg/dL, with Group 1 at 54 ± 34.4 mg/dL and Group 2 at 88 ± 53 mg/dL; however, this difference was not statistically significant (*p* = 0.09).

Regarding K levels, the mean at admission was 4.5 ± 0.56 mmol/L overall, with Group 1 at 4.45 ± 0.54 mmol/L and Group 2 at 4.72 ± 0.61 mmol/L (*p* = 0.57). By discharge, the overall K levels were 4.17 ± 0.55 mmol/L, with Group 1 showing 4.3 ± 0.56 mmol/L and Group 2 demonstrating lower levels at 3.89 ± 0.54 mmol/L (*p* = 0.66).

Sodium levels remained similar between the groups, with an overall mean of 139 ± 3.80 mmol/L at admission; Group 1 had a mean of 139 ± 3.56 mmol/L, while Group 2 had 139 ± 4.11 mmol/L (*p* = 0.37). At discharge, Na levels were recorded as 138 ± 13.1 mmol/L overall, with Group 1 at 138 ± 16.61 mmol/L and Group 2 at 137 ± 3.48 mmol/L (*p* = 0.60).

### 3.5. ∆ePVS and NT-proBNP

The median NT-proBNP level at admission was 4060 pg/dL overall, with Group 1 showing a median of 3626 pg/dL and Group 2 showing 4855.5 pg/dL, a statistically significant difference (*p* = 0.005). At discharge, the overall median NT-proBNP level decreased to 1561.5 pg/dL, with Group 1 at 1454 pg/dL and Group 2 at 1899 pg/dL (*p* = 0.01).

The association between ∆ePVS and NT-proBNP indicated a nearly threefold increased risk (OR 2.96, 95% CI 1.61–5.43, *p* < 0.008), suggesting a significant correlation.

Figure 2 illustrates the relationship between ∆ePVS and the proportion of patients who achieved a reduction in NT-proBNP ≥ 30% or more. Patients in Group 1 had a significantly higher likelihood of achieving this reduction (88.9%) compared to those in Group 2, where only 11.1% showed similar results.

The ROC curve analysis, evaluating the relationship between NT-proBNP reduction and ∆ePVS, demonstrated a moderate predictive value with an AUC of 0.682 (95% CI 0.533–0.830, *p* = 0.008) (Figure 3).

The multivariate analysis was conducted using multivariate logistic regression model. The independent variables included in the model were LVEF, IVC diameter at discharge, sPAP at discharge, E/e’ ratio at discharge, and ∆ePVS. The variables were selected based on their potential association with NT-proBNP reduction. The reduction in NT-proBNP of >30% (used in dichotomous form) at discharge was used as the reference point for assessing decongestion and served as the dependent variable in the regression analysis. The analysis identified ∆ePVS as the only significant predictor of NT-proBNP reduction at discharge (OR 0.11, 95% CI 0.04–0.33, *p* < 0.001).

### 3.6. ∆ePVS and Clinical Outcomes

The absence of ∆ePVS was linked to a higher risk of in-hospital mortality (OR 1.30, 95% CI 1.07–1.58, *p* < 0.01) and six-month mortality (OR 1.30, 95% CI 1.02–1.66, *p* = 0.04). However, it did not show statistical significance in predicting rehospitalization (OR 1.25, 95% CI 0.82–1.93, *p* = 0.25). During hospitalization, five patients in Group 2 died. At the six-month follow-up, the rehospitalization rate was 35% in Group 2 (10 patients) and 21% in Group 1 (17 patients) (*p* = 0.04). Mortality rates at six months were 37% in Group 2 (11 patients) and 11% in Group 1 (8 patients) (*p* = 0.012).

## 4. Discussion

It has been proposed that changes in easily accessible parameters, such as Hb and Ht, could serve as surrogates for effective decongestion, aiding in the assessment of therapeutic efficacy in patients with ADHF. This was supported by several large retrospective post hoc studies that demonstrated an association between hemoconcentration (indicated by increased Hb concentrations or Ht levels) and improved clinical outcomes in hospitalized ADHF patients [11].

We analyzed 111 patients with ADHF and, using the Strauss formula, divided them based on ∆ePVS. Subsequently, we assessed clinical, biochemical, and echocardiographic parameters.

Patients in Group 2, who had worse decongestion outcomes, presented with poorer clinical status upon admission compared to Group 1. Specifically, although the weight at admission showed no significant differences between groups, Group 2 had a higher prevalence of lower-limb edema and jugular vein distension, which are indicative of more severe congestion. These clinical findings may reflect the greater diuretic requirement observed in Group 2.

In our study, a borderline statistically significant difference in weight loss was observed between the groups, with Group 1 showing a median weight difference of 2 kg (IQR: 1–3) compared to 1 kg (IQR: 0–2) in Group 2 (*p* = 0.046). This result highlights the limitations of weight changes as a surrogate for decongestion. While weight loss may reflect clinical decongestion, it does not reliably indicate changes in intravascular volume or predict better ADHF outcomes. Patient-to-patient variability in volume status at admission and discharge further complicates its interpretation [18].

No statistically significant differences were identified between the groups regarding age, sex, BMI, BP, HR, or smoking status. There were also no statistically significant differences in comorbidities between the two groups. Group 2 patients received significantly fewer MRAs at admission, and this difference is likely attributable to the more severe renal dysfunction present at admission in this group.

There were also no significant differences regarding SGLT-2 inhibitors between the two groups despite the evidence suggesting that treatment with SGLT-2 inhibitors leads to a 7.3% reduction in measured plasma volume from baseline, with consistent increases in Ht and Hb levels [19]. This effect is likely more relevant in patients with chronic heart failure rather than acute cases.

Among the clinical parameters for assessing congestion, no statistically significant differences were observed between the two groups, suggesting that the clinical congestion was similar. Regarding echocardiographic data, no statistically significant differences in LVEF were found between the groups. Previous studies have shown that congestion levels are similar regardless of LVEF category [20].

Regarding renal function, Group 2 exhibited higher baseline creatinine levels (1.34 ± 0.57 mg/dL vs. 1.08 ± 0.49 mg/dL in Group 1), likely due to more severe comorbidities such as diabetes and hypertension. These conditions are associated with an increased risk of diuretic resistance, which contributes to poorer outcomes in acute decompensated heart failure.

The analysis of renal function demonstrated that Group 2 patients with WRF had significantly higher in-hospital mortality and six-month mortality compared to Group 1 patients with WRF. These results support prior findings indicating that hemoconcentration, although associated with WRF, does not adversely affect prognosis [21,22,23]. On the other hand, WRF occurring without hemoconcentration, particularly in the presence of persistent congestion, is associated with significantly poorer outcomes [24,25].

In this study, the significant difference in NT-proBNP levels between Group 1 and Group 2 at admission and discharge highlights the potential role of this biomarker in distinguishing different patient risk profiles. The findings align with the previous work, which similarly identified elevated NT-proBNP levels as a marker for worse prognosis in heart failure patients [15]. The higher baseline and discharge levels in Group 2 suggest a more severe disease state and potentially poorer response to treatment, as reflected by the lower proportion of patients achieving a ≥30% NT-proBNP reduction.

The correlation between plasma ∆ePVS and NT-proBNP reduction supports the hypothesis that effective decongestion is a key factor in improving patient outcomes. The nearly threefold increased risk associated with ∆ePVS indicates that this parameter could be a valuable addition to the clinical assessment of ADHF patients, particularly in settings where biomarkers like NT-proBNP are not routinely available due to cost constraints.

The ROC curve analysis further reinforces the moderate predictive value of ∆ePVS for NT-proBNP reduction. Its value lies in complementing other biomarkers and clinical assessments. It provides a simple, dynamic measure of intravascular volume changes, aiding in evaluating decongestion therapy, especially in resource-limited settings. This indicator is best utilized as part of a multimodal approach, integrating clinical and biochemical parameters to enhance patient management.

In this study, multivariate regression analysis identified ∆ePVS as the sole significant predictor of NT-proBNP decrease. This emphasizes the importance of monitoring and managing plasma volume, especially since it can be derived from routine blood tests without incurring additional costs. This aspect becomes increasingly relevant in healthcare systems facing resource limitations, as highlighted by the literature [26,27].

In conclusion, while NT-proBNP remains a valuable biomarker for assessing prognosis and guiding therapy in ADHF patients, ∆ePVS offers a cost-effective alternative for evaluating decongestion and predicting outcomes. Further studies are needed to validate these findings and explore the integration of plasma volume assessment into standard clinical practice for ADHF management.

Our study demonstrated that the absence of ∆ePVS was significantly associated with an increased risk of both in-hospital mortality and six-month mortality. However, it did not show a statistically significant correlation with the risk of rehospitalization (OR 1.25, 95% CI 0.82–1.93, *p* = 0.25). These findings are consistent with the previous literature. For instance, a study involving 712 patients with ADHF observed that an increase in Strauss-derived estimated plasma volume at discharge was associated with a worse prognosis [28]. Similarly, changes in ∆ePVS have been identified as prognostic indicators for death, ADHF hospitalization, and all-cause mortality in patients with ADHF, as shown in an observational cohort study [9]. Additional evidence from a post hoc analysis of the EVEREST study highlighted that hemoconcentration correlated with improved congestion and reduced mortality in such patients [29].

The exact mechanism through which plasma volume changes influence prognosis in ADHF remains uncertain. However, some studies have challenged the use of ∆ePVS as a reliable clinical indicator. For instance, a study involving 36 patients with congestion monitored estimated and directly measured plasma volume changes, revealing that variations in Hb and Ht did not accurately correspond to the actual changes in plasma volume measured in ADHF patients [11]. These findings indicate that relying on Hb or Hct levels alone may not provide a precise assessment of intravascular volume status in patients with ADHF.

To date, the exact mechanism through which changes in plasma volume impact prognosis remains unclear [30,31].

## 5. Limitations

Our study has several limitations that should be acknowledged. First, it was a prospective observational cohort study conducted at a single center. The sample size was relatively small, and we included only patients whose primary diagnosis was ADHF, excluding those with ADHF secondary to other conditions. This limits the applicability of ∆ePVS to a broader population of ADHF patients. Furthermore, we were unable to directly measure plasma volume for comparison, relying instead on estimates.

A limitation of our study is the lack of direct quantification of the volume removed at the second plasma volume measurement. While ∆ePVS reflects changes in plasma volume, measuring the removed volume could provide further insights into decongestion dynamics.

Lastly, the unequal distribution of patients between the groups may have impacted the results and limited the strength of the conclusions drawn.

While targeting ∆ePVS < 0% may guide therapy, it should not be an absolute goal for all patients. This metric is best used alongside clinical assessment to individualize treatment and optimize outcomes.

## 6. Conclusions

Indirectly ∆ePVS is a cost-effective and accessible parameter for evaluating of decongestive treatment in ADHF patients. Unlike natriuretic peptides, ∆ePVS can be derived from routine blood tests, making it an affordable tool that could be integrated into clinical practice.

Our study suggests that ∆ePVS may predict mortality in ADHF patients post discharge. The simplicity and low cost of this parameter make it a promising tool for risk stratification and targeted treatment. However, further research is necessary to validate its utility in larger and more diverse cohorts of heart failure patients.

## Figures and Tables

**Figure 1 biomedicines-13-00088-f001:**
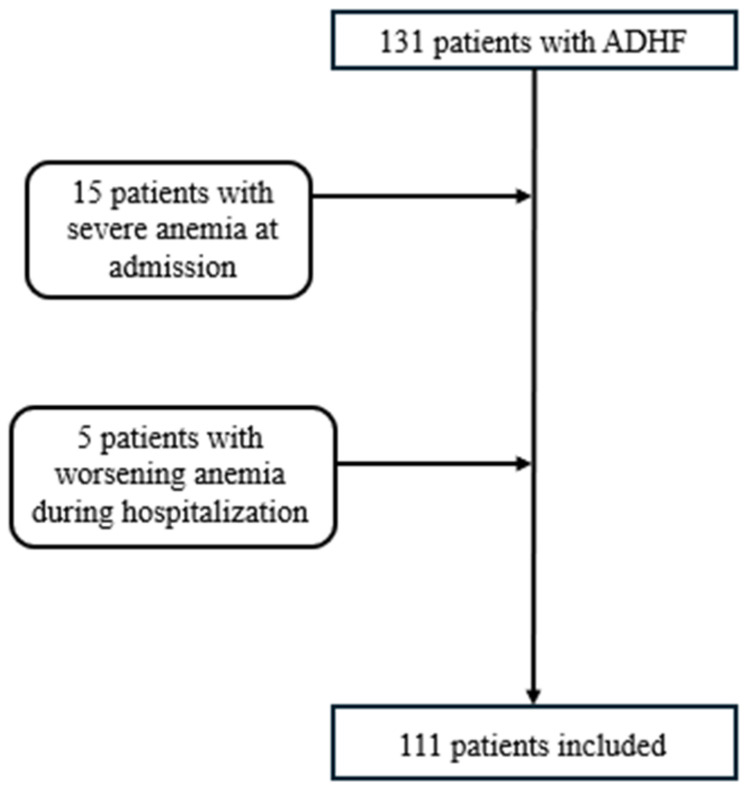
Patient Selection Flowchart.

**Figure 2 biomedicines-13-00088-f002:**
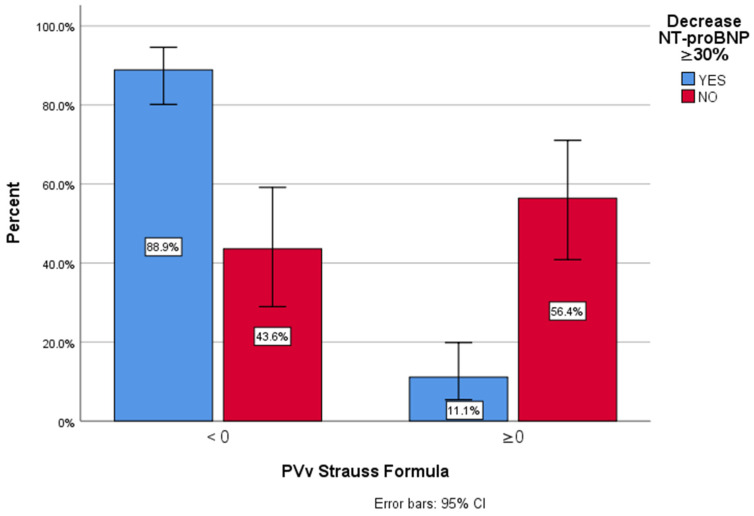
Correlation Between ∆ePVS and NT-proBNP reduction at discharge (≥30%).

**Figure 3 biomedicines-13-00088-f003:**
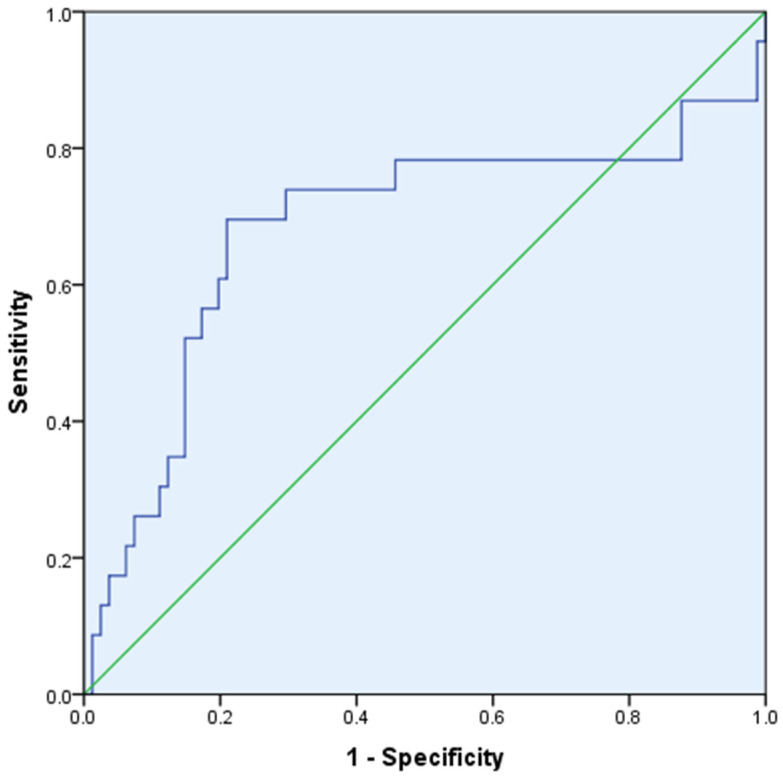
ROC Curve Analysis of NT-proBNP Reduction and ∆ePVS Predictive Value (AUC 0.682, *p* = 0.008).

**Table 1 biomedicines-13-00088-t001:** Clinical and demographic characteristics of both groups.

Baseline Admission Characteristics	Total (111 Patients)	Groups		
		Group 1 81 Patients	Group 2 30 Patients	*p*-Value
Age, years, median (IQR)	74 (67–82)	73 (66–82)	74 (70–85)	0.20
Sex, female (%)	39.29	40.7	36.67	0.56
BMI (kg/m^2^), median (IQR)	28.71 (24–33)	28.57 (24–45)	30.51 (27–32)	0.23
HR, beats/min, median (IQR)	76 (69–100)	75 (70–100)	76.5 (66–100)	0.66
SBP, mmHg, median (IQR)	130 (110–149)	130 (115–147)	130 (110–150)	0.68
Smoker (%)	41.08	41.98	40	0.59
Comorbidities				
IHD, (%)	39.29	44.44	26.67	0.14
HTN, (%)	82.14	81.48	86.67	0.49
DM (%)	36.61	39.51	30	0.39
CKD (%)	60.71	56.79	73.33	0.16
AF (%)	57.14	54.32	66.67	0.30
COPD (%)	22.32	20.99	26.67	0.49
Medication at admission				
ACEi (%)	33.04	32.10	36.67	0.54
ARB (%)	16.07	18.52	10	0.33
ARNI (%)	17.86	20.99	10	0.24
Beta-blocker (%)	83.93	90.12	70	0.06
MRA (%)	68.75	79.01	43.33	0.0008
SGLT-2 inhibitors (%)	34.82	37.04	30	0.47
CRT (%)	5.40	5	1	0.08

BMI—body mass index; HR—heart rate; SBP—systolic blood pressure; IHD—ischemic heart disease; HTN—hypertension; DM—diabetes mellitus; CKD—chronic kidney disease; AF—atrial fibrillation; COPD—chronic obstructive pulmonary disease; ACEi—angiotensin-converting-enzyme inhibitors; ARB—angiotensin II receptor blockers; ARNI—angiotensin receptor receptor-neprilysin inhibitor; MRA—mineralocorticoid receptor antagonists; CRT—cardiac resynchronization therapy.

**Table 2 biomedicines-13-00088-t002:** Clinical Congestion Parameters at Admission and Discharge.

Parameter	Total (111 Patients)	Group 1 (81 Patients)	Group 2 (30 Patients)	*p*-Value
Weight (kg)				
Admission, mean	83 ± 18.02	83.5 ± 18.82	83.8 ± 15.95	0.92
Discharge, mean	82 ± 17.10	80 ± 17.77	82 ± 14.55	0.17
Weight difference, median, IQR	2 (0–3)	2 (1–3)	1 (0–2)	0.046
Lower-limb edema (%)				
Admission	36	31	48	0.28
Discharge	9	7	12	0.15
Jugular vein distention (%)				
Admission	71	68	75	0.38
Discharge	13	11	15	0.29

**Table 3 biomedicines-13-00088-t003:** Echocardiographic and Lung Ultrasound Findings at admission.

Parameter	Total 111 Patients	Group 1 (81 Patients)	Group 2 (31 Patients)	*p*-Value
Echocardiography				
LVEF, mean	46.74 ± 15.08	46.19 ± 15.16	46.03 ± 12.77	0.95
E/e’ medial ratio, median	28 (20–37)	28.33 (19–39)	29 (21–32)	0.31
LAd (mm), mean	47 ± 6.99	47 ± 5.8	48 ± 8.9	0.87
TAPSE (mm), mean	17 ± 3.9	17 ± 3.8	16 ± 4.2	0.50
S′RV (cm/s), mean	11 ± 2	11.73 ± 1.9	11.31 ± 2.5	0.49
IVC (mm), mean	22 ± 4.8	22 ± 4.6	21 ± 5.3	0.82
sPAP (mm Hg), mean	49 ± 14.1	49 ± 13	50 ± 16	0.36
Lung Ultrasound				
Pleural fluid (%)	48.65	46.91	53.33	0.45
Signs of pulmonary congestion (%)	87.39	86.42	90	0.54

LVEF—left-ventricular ejection fraction; LAd—left atrium diameter; TAPSE—tricuspid annular plane systolic excursion; S′RV—S′ right ventricle; IVC—inferior vena cava; sPAP—systolic pulmonary arterial pressure.

**Table 4 biomedicines-13-00088-t004:** Biochemical Parameters at Admission and Discharge and Diuretic Treatment at admission and during hospitalization.

Baseline	Total (111 Patients)	Group 1 (81 Patients)	Group 2 (30 Patients)	*p*-Value	Total 111 Patients	Group 1 (81 Patients)	Group 2 (30 Patients)	*p*-Value
Time	Admission				Discharge			
Biochemical								
Hb (g/dL), mean	12 ± 2.66	12.3 ± 2.68	11.75 ± 2.63	0.90	12.7 ± 2.4	13.4 ±2.34	11.2 ± 2.74	0.33
Ht (%)	37.4 ± 7.7	37.34 ± 7.62	37.8 ± 7.82	0.92	39.1 ± 7.02	40.6 ± 6.79	34.5 ± 9.01	0.33
Creatinine (mg/dL), mean	1.16 ± 0.65	1.08 ± 0.49	1.34 ± 0.57	0.048	1.19 ± 0.43	1.13 ± 0.49	1.32 ± 0.61	0.00418
eGFR, (1.73 mL/min/mp), median	54.50 (41–71)	56 (43–73)	51 (31–61)	0.89	58.89 (44–73)	59.67 (45–74)	55.86 (40–64)	0.86
WRF (%)					42.34	39.51	50	0.27
Serum urea (mg/dL)	56 ± 38.5	54 ± 30	73 ± 55	0.0051	59 ± 37.1	54 ± 34.4	88 ± 53	0.09
K (mmol/L)	4.5 ± 0.56	4.45 ± 0.54	4.72 ± 0.61	0.57	4.17 ± 0.55	4.3 ± 0.56	3.89 ± 0.54	0.66
Na (mmol/L)	139 ± 3.80	139 ± 3.56	139 ± 4.11	0.37	138 ± 13.1	138 ± 16.61	137 ±3.48	0.60
NT-proBNP (pg/dL), median	4060 (1865–7085)	3626 (2456–6254)	4855.5 (1203–14,243)	0.005	1561.5 (1013–2921)	1454 (1002–2741)	1899 (1120–5242)	0.01
Diuretic dose at admission and at discharge								
Furosemide (mg/day), median (IQR)	20 (0–40)	20 (0–35)	20 (0–40)	0.06	60 (40–80)	60 (40–80)	100 (40–120)	0.03
Diuretic dose during hospitalization								
Furosemide (mg/day), median (IQR)	85 (60–110)	75 (50–90)	95 (70–120)	0.03				

Hb—hemoglobin; Ht—hematocrit; eGFR—estimated glomerular filtration rate; WRF—worsening renal function; K—potassium; Na—sodium.

## Data Availability

Data used in this study may be provided by the authors upon reasonable request.

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
