# Peer review of "The Role of the Estimated Plasma Volume Variation in Assessing Decongestion in Patients with Acute Decompensated Heart Failure"

_biomedicines, 2025, doi:10.3390/biomedicines13010088_

Round 1
Reviewer 1 Report
Comments and Suggestions for Authors
I enjoyed reading your paper. I have a few questions.
1. How would one use this method of measuring plasma volume in relationship to treatment of the patient. Should one keep diuresing until the equation demonstrates a value <0%?
2. Throughout the paper you use the abbreviation ADHF and HF, do yuo consider the definition the same? I would think you would want to use HDHF throughout.
3. In the follow-up section. DId you use the same criteria for "HF" as you used for ADHF in the inclusion criteria
4. Line 162. i would add "at discharge" we divided
5. It is intersesting that the wt loss was not significant between groups. Can you explain
6. It might be interesting to look at amount of volume removed at 2nd measurement of plasma volume
7. How did you evaluate/measure limb edema.
8. It appears that overal clinical changes did not make a difference in readmission. I find that very interesting. What was the criteria for discharge
Author Response
Response to Reviewer 1 Comments
Thank you for your thorough review and thoughtful comments. We have addressed your questions and suggestions in detail below, and the corresponding changes have been implemented in the manuscript. These changes are highlighted in red in the revised document.
- How would one use this method of measuring plasma volume in relation to treatment of the patient? Should one keep diuresing until the equation demonstrates a value <0%?
Response 1:
Thank you for this insightful question. Based on your suggestion, we have clarified in the discussion section that while a negative ∆ePVS (<0%) indicates hemoconcentration and effective decongestion, it should not be treated as an absolute goal for all patients. Clinical judgment remains crucial, taking into account symptoms, clinical signs, and potential risks like renal dysfunction or electrolyte imbalances.
The revised Limitations section (page 13, lines 440-442) now states: “While targeting ∆ePVS <0% may guide therapy, it should not be an absolute goal for all patients. This metric is best used alongside clinical assessment to individualize treatment and optimize outcomes.”
This adjustment aims to reflect a more balanced interpretation of the metric, as highlighted in your feedback.
- Throughout the paper, you use the abbreviations ADHF and HF. Do you consider the definitions the same? I would think you would want to use ADHF throughout.
Response 2:
Thank you for your insightful observation. ADHF and HF were initially used to distinguish acute decompensated heart failure from chronic heart failure. However, we recognize that this may have caused confusion. Based on your feedback, we have revised the manuscript to consistently use "ADHF" when referring to the acute setting. This adjustment has been applied throughout the text to ensure clarity (e.g., page 12, line 387).
Your comment has been very helpful in improving the consistency and precision of the terminology used in the manuscript.
- In the follow-up section, did you use the same criteria for "HF" as you used for ADHF in the inclusion criteria?
Response 3:
Thank you for your valuable question. We appreciate the opportunity to clarify this point. Yes, we used the same criteria for "HF" in the follow-up section as we did for ADHF in the inclusion criteria. Specifically, ADHF hospitalization was defined as an admission where heart failure (HF) was the primary diagnosis, characterized by symptoms and signs of HF that required treatment with intravenous diuretics.
To ensure better clarity and consistency, we have also revised the manuscript to use "ADHF" throughout, including in the follow-up section, instead of the more general "HF." This change helps maintain uniformity and avoid confusion.
We have added the following clarification to the manuscript:
" We used the same criteria for "HF" as we did for ADHF in the inclusion criteria." - (page 3, lines 144-145).
We hope this clarification addresses your concern, and we appreciate your attention to detail.
- Line 162: I would add "at discharge" to "we divided."
Response 4:
Thank you for this suggestion. We have added "at discharge" to clarify the statement. The updated sentence now reads: "[ A total of 111 patients were divided at discharge based on ∆ePVS into two groups: Group 1 with 81 patients and Group 2 with 30 patients.]." This revision is located on page 4, line 175. - It is interesting that the weight loss was not significant between groups. Can you explain?
Thank you for this observation. After recalculating the weight loss, we found a borderline statistically significant difference between the two groups, with Group 1 showing a median weight difference of 2 kg (IQR: 1–3) compared to 1 kg (IQR: 0–2) in Group 2 (p = 0.046). This updated finding has been included in the revised results section (Page 6, Table 2, line 215).
While weight loss is often used as a surrogate for clinical decongestion in ADHF patients, its variability limits its reliability as an indicator of intravascular volume changes. Recent evidence suggests that diuresis-related changes in weight may not consistently correlate with improved heart failure outcomes and do not accurately reflect changes in intravascular volume (Reference: Wayne L. Miller, Ronstan Lobo, Why is Diuresis-related Weight Loss Not a Consistent Predictor of Clinical Outcomes in Heart Failure? -need to Account For Variability in Volume Status, Journal of Cardiac Failure).
To address your comment, we have updated the discussion as follows (page 11, lines 343–349):
„In our study, a borderline statistically significant difference in weight loss was observed between the groups, with Group 1 showing a greater median weight difference of 2 kg (IQR: 1–3) compared to 1 kg (IQR: 0–2) in Group 2 (p = 0.046). While weight loss may reflect clinical decongestion, it does not reliably indicate changes in intravascular volume or predict better outcomes in ADHF. Variability in volume status at admission and discharge complicates the interpretation of weight changes as a surrogate marker of decongestion. This finding highlights the limitations of relying solely on weight changes to evaluate decongestion in ADHF patients.”
We appreciate this valuable comment, which allowed us to refine our analysis and incorporate further insights into the discussion of clinical decongestion.
- It might be interesting to look at the amount of volume removed at the 2nd measurement of plasma volume.
Response 6:
We agree that this analysis could provide additional insights. However, the study did not specifically quantify the volume removed beyond ∆ePVS values. This would indeed be very interesting and could yield valuable data. We are considering incorporating this aspect in future research to develop a larger and more comprehensive study. This limitation has been acknowledged in the Limitation (page 13, lines 434-437).
- How did you evaluate/measure limb edema?
Response 7:
Thank you for raising this important point. Lower limb edema was evaluated clinically by applying digital pressure to assess for pitting and swelling. This method was consistently performed during clinical evaluations at admission to ensure uniformity. The description of this evaluation can be found in the Methods section - Clinical evaluations were performed at admission, lower limb edema was assessed clinically by applying digital pressure to evaluate pitting and swelling (page 3, lines 96-97). We appreciate the opportunity to clarify this aspect of our methodology.
- It appears that overall clinical changes did not make a difference in readmission. I find that very interesting. What was the criterion for discharge?
Response 8:
Thank you for this observation. Discharge was determined based on overall clinical stability, which included the resolution of symptoms such as orthopnea and congestion-related complaints, alongside improvements in laboratory markers and echocardiographic parameters. Importantly, the final decision for discharge was made by the attending physician, resulting in some variability due to the lack of a fully standardized protocol. This information has been added to the Methods section for clarity (page 3, lines 136-138).
Reviewer 2 Report
Comments and Suggestions for Authors
This study aims to evaluate estimating plasma variation (∆ePVS) for evaluating decongestion in patients with acute decompensated heart failure (ADHF).
The authors need to address the following questions:
- Clarify the reason for selecting this indicator.
- Given that the area under the ROC curve is <0.7, what are the diagnostic advantages of this indicator? Does it offer earlier prediction of disease progression/prognosis or greater accuracy?
Author Response
Response to Reviewer 2 Comments
Thank you for your thoughtful review and constructive feedback. Below, we provide detailed responses to your comments and have made corresponding revisions in the manuscript, which are highlighted in red in the revised submission.
- Clarify the reason for selecting this indicator.
Response 1:
Thank you for pointing this out. The selection of ∆ePVS was based on its validated use as a surrogate for plasma volume changes, as demonstrated in prior studies using the Strauss formula. This indicator provides a non-invasive, cost-effective, and easily accessible means of assessing decongestion, using parameters that are routinely measured in clinical practice (hemoglobin and hematocrit). Unlike natriuretic peptides, which may be limited by cost or availability, ∆ePVS can be applied universally without additional resources. This has been further clarified in the Methods sections (page 3, lines 118-120).
- Given that the area under the ROC curve is <0.7, what are the diagnostic advantages of this indicator? Does it offer earlier prediction of disease progression/prognosis or greater accuracy?
Response 2:
We acknowledge the modest AUC of 0.682, indicating that ∆ePVS alone has moderate predictive power. However, its advantages lie in its ability to complement existing biomarkers and clinical assessments. ∆ePVS offers a simple, dynamic measure of intravascular volume status that reflects the effectiveness of decongestion therapy. While it may not provide earlier prediction compared to natriuretic peptides or imaging, it is a valuable adjunct, particularly in resource-limited settings. This point has been elaborated in the Discussion section (page 12, lines 389-394).
Furthermore, we discuss its utility as part of a multimodal approach, emphasizing that its strength lies in accessibility and integration with clinical and biochemical parameters. This revision aims to contextualize the moderate AUC and highlight the practical applicability of ∆ePVS.
Reviewer 3 Report
Comments and Suggestions for Authors
The paper is interesting and raises a clinically important issue.
I have few comments and questions:
1) the paper needs some editorial correction:
- The sentence: „Several formulas are available for estimating plasma volume, but the Strauss formula was chosen because it has been validated against a radiolabeled gold standard method and has been consistently used for decades to estimate plasma volume in patients undergoing plasma exchanges” is doubled.
- Spelling mistakes: hemoglobi, hematocri
- no expansion of the abbreviation: S’, a, d (admission, discharge – in the equation in the abstract)
- It is unclear: “The analysis of renal function highlighted a significant difference in outcomes for patients in Group 2 who developed WRF. Compared to Group 1 patients with WRF, those in Group 2 had a significantly higher risk of in-hospital mortality (OR 1.66, 95% CI 1.10- 2.51, p=0.02) and six-month mortality (OR 1.87, 95% CI 1.16-3.01, p<0.01). These findings align with previous studies that have shown that hemoconcentration is often associated with WRF but does not negatively impact prognosis [21-23].” Please rephrase
- The discussion reiterates the results
- unnecessarily in the section describing the results are rewritten data from the tables
2) Why the Friedman test was applied?
3) How was the worsening of anemia, which was an exclusion criterion, defined?
4) The weight difference between admission and discharge in the two groups should be compared, not the values at the two time points
5) there is no information on what data the model for multivariate analysis is built from
6) There is no information on the treatment given and the monitoring of the response in both groups
7) the fact that the groups differed in their baseline renal function is under-emphasised. This is commented on in the context of the NT-pro-BNP results, but should be discussed in the broader context of differences in the clinical status of patients on admission
Round 2
Reviewer 3 Report
Comments and Suggestions for Authors
Thank you for all the corrections and additional information.
There are still issues that have not been addressed:
1) the paper needs some editorial correction:
- no expansion of the abbreviation: S’
2) there is still no information on what exact data the model for multivariate analysis is built from – the model should be presented
3) There is no information on the dose of diuretics given during hospitalization (it could be calculated for day o hospitalization); table presents only information about diuretics at the admission and at the discharge
4) There is still a lack of discussion about the poorer clinical status on admission of patients with worse decongestion.
Author Response
Thank you for your review and thoughtful comments. We have addressed your questions and suggestions in detail below, and the corresponding changes have been implemented in the manuscript. These changes are highlighted in red in the revised document.
- The paper needs some editorial correction: no expansion of the abbreviation: S’.
Response 1:
Thank you for pointing out the need for clarification regarding the abbreviation "S’". We have now ensured that the abbreviation is properly defined as lateral tricuspid annulus peak systolic velocity (S’) in the Materials and Methods section (line 127). The correction has been implemented and is highlighted in the revised manuscript.
- There is still no information on what exact data the model for multivariate analysis is built from – the model should be presented
Response 2:
Thank you for your observation. We appreciate the opportunity to clarify the multivariate analysis model. In the revised manuscript, we have now provided additional details about the model used. The multivariate analysis was conducted using multivariate logistic regression model. The independent variables included in the model were: LVEF, IVC diameter at discharge, sPAP at discharge, E/e' ratio at discharge and ∆ePVS. The variables were selected based on their potential association with NT-proBNP reduction. The reduction of NT-proBNP by >30% (used in dichotomous form) at discharge was used as the reference point for assessing decongestion and served as the dependent variable in the regression analysis.. The analysis identified ∆ePVS as the only significant predictor of NT-proBNP reduction at discharge (OR 0.11, 95% CI 0.04–0.33, p<0.001).
The correction has been implemented and is highlighted in the revised manuscript (lines 329-336).
- There is no information on the dose of diuretics given during hospitalization (it could be calculated for day of hospitalization); table presents only information about diuretics at the admission and at the discharge.
Response 3:
Thank you for your observation. We appreciate the opportunity to address this point. We have collected data on the total diuretic dose administered during hospitalization for each group, and we have now included this information in both the table and the results section of the revised manuscript.
The total diuretic dose during hospitalization was higher in Group 2 compared to Group 1, reflecting a greater need for diuretic treatment in Group 2.
The correction can be found in Table 4 and in the Results section, lines 280 – 286 and Discussion section, lines 359-360 of the revised manuscript.
- There is still a lack of discussion about the poorer clinical status on admission of patients with worse decongestion.
Response 4:
Thank you for your observation. We appreciate the opportunity to address this important point. In response, we have added a discussion on the poorer clinical status of patients in Group 2 at admission, which may have contributed to their worse decongestion outcomes. Specifically, Group 2 had a higher prevalence of lower limb edema and jugular vein distension at admission, suggesting more severe congestion compared to Group 1. These clinical parameters were highlighted in the Results section, and we have now discussed their potential impact on decongestion in the revised Discussion section (lines 355-359 of the revised manuscript).